# The Role of Noninvasive ^13^C-Octanoate Breath Test in Assessing the Diagnosis of Nonalcoholic Steatohepatitis

**DOI:** 10.3390/diagnostics12122935

**Published:** 2022-11-24

**Authors:** Carmen Fierbinteanu-Braticevici, Ana-Maria Calin-Necula, Vlad-Teodor Enciu, Loredana Goran, Anca Pantea Stoian, Ioan Ancuta, Octav Viasu, Alexandru Constantin Moldoveanu

**Affiliations:** 1Department of Gastroenterology, “Carol Davila” University of Medicine and Pharmacy, 050474 Bucharest, Romania; 2Emergency University Hospital, 050098 Bucharest, Romania; 3Department of Diabetes, Nutrition and Metabolic Diseases, “Carol Davila” University of Medicine and Pharmacy, 050474 Bucharest, Romania; 4“Prof. Dr. N. C. Paulescu” National Institute of Diabetes, Nutrition and Metabolic Diseases, 030167 Bucharest, Romania; 5Internal Medicine Department, Dr. I. Cantacuzino Hospital, 030167 Bucharest, Romania

**Keywords:** fatty liver, non-alcoholic fatty liver disease, breath tests, diagnostic techniques, digestive system

## Abstract

Background: The diagnosis of NASH needs a liver biopsy, an invasive procedure that is not frequently accepted by patients. The aim of our study was to evaluate the efficacy of the ^13^C-Octanoate breath test (OBT) as a non-invasive surrogate marker to differentiate patients with NASH from patients with simple steatosis (NAFL). Methods: We performed a prospective study on patients with histologically established non-alcoholic steatohepatitis and no other hepatic disease. Each patient underwent a testing protocol, which included a clinical exam, laboratory blood tests, standard abdominal ultrasound, and a ^13^C-Octanoate breath test. Results: The study group included: 82 patients with steatohepatitis, 64 patients with simple steatosis, and 21 healthy volunteers. The univariate and bivariate analysis identified that significant values were the percent dose recovery (PDR) at 15 min—r = 0.65 (AUROC = 0.902) and cumulative percent dose recovery (cPDR) at 120 min—r = 0.69 (AUROC = 0.899). Discussion: Our study showed that ^13^C-OBT had good efficacy for identifying patients with NASH from those with NAFL (steatosis alone) but not those with NAFL from healthy subjects. Considering all these pathogenic steps in NASH we considered that OBT could have the clinical utility to identify patients at risk for NASH, especially “fast progressors”.

## 1. Introduction

Non-alcoholic fatty liver disease (NAFLD) is the liver manifestation of the metabolic syndrome and includes a large spectrum of clinicopathological conditions: nonalcoholic fatty liver (NAFL), nonalcoholic steatohepatitis (NASH), fibrosis, and cirrhosis. Characterized by more than 5% hepatic fat accumulation and the exclusion of other causes of liver steatosis, NAFLD is becoming a major cause of liver disease-related mortality. Its estimated prevalence has reached pandemic proportions, affecting one-third of the general population [1]. Recently, a group of experts concluded that the diversified pathogenesis paths of NAFLD and imprecisions in terminology and definitions required a new nomenclature that expresses the current state of knowledge more accurately. Metabolic dysfunction-associated fatty liver disease (MAFLD) was proposed as a better term [2].

Emerging data reports that NAFLD subtypes have a different potential for disease progression and overall mortality. Several large studies confirm that individuals with histologic-proven NASH are at a higher risk of disease progression and liver-related mortality than patients with non-NASH subtypes [3,4]. In fact, NASH, which is defined by liver steatosis and inflammation with hepatocyte injury, is accepted to be the only entity in the clinical spectrum of NAFLD which possesses a high risk of evolution toward end-stage liver disease and overall mortality. Moreover, NASH promotes liver carcinogenesis due to genetic alterations and liver-specific molecular mechanisms, which is an independent risk factor for the development of NASH-associated hepatocellular carcinoma (HCC), even in a non-cirrhotic liver [5].

The correct diagnosis of NASH involves liver biopsy, the only procedure that can differentiate between NAFLD and NASH [6]. However, this is not frequently accepted by patients, and a large number of studies show a significant sampling variability between intra- and inter-observers [7]. As this histologic confirmation bias can result in staging and even diagnostic errors in a high proportion of patients, the potential effectiveness of non-invasive tests that are able to identify individuals at high risk for disease progression is investigated.

A key component in NASH pathophysiology is the altered hepatic beta-oxidation. The accumulation of liver fat with a subsequently increased hepatic uptake and synthesis of free fatty acids (FFA) is balanced by an increased mitochondrial beta-oxidation and ketogenesis [8]. In light of these data, the non-invasive assessment of hepatic fatty acid metabolism might provide more insight into the progression of the fatty liver to steatohepatitis and treatment monitoring.

Forty years ago, ^13^C-based breath tests were applied for the non-invasive investigation of liver mitochondrial function [9]. A method with good practicability in the non-invasive investigation of liver functions can be considered the ^13^C-octanoic acid breath test. Octanoic acid is a medium-chain fatty acid composed of eight carbon atoms that were initially validated to be used in a non-invasive breath test to measure the gastric emptying rate of solids [10]. Octanoic acid has the advantage of being rapidly absorbed in the small intestine and transported through the portal vein in the liver, where it is metabolized by beta-oxidation in acetyl-CoA and CO_2_. In the end, CO_2_ is exhaled by the subject and collected in breath samples at different time points. ^13^C isotope was chosen as it is the only stable isotope other than ^12^C [11].

The aim of our study was to evaluate the efficacy of the ^13^C-Octanoate breath test (OBT), as a surrogate marker of the mitochondrial function, in differentiating patients with NASH from patients with simple steatosis (NAFL).

## 2. Materials and Methods

We performed a prospective study on patients with non-alcoholic steatohepatitis and no other hepatic disease between October 2014 and September 2017. Each patient underwent a testing protocol, which included a clinical exam, laboratory blood tests, standard abdominal ultrasound, and a ^13^C-Octanoate breath test. The definitive diagnosis of non-alcoholic steatohepatitis was established using hepatic biopsy. All tests were performed within a maximum of 72 h of each other, with the notable exception of the hepatic biopsy, which was performed within the previous 6-months. The study was approved by the Local Ethics Committee (nr. 22052/11 May 2015), and an informed consent form was signed by each patient.

The main Inclusion criteria consisted of patients at least 18 years of age, with previously established or newly established diagnosis of non-alcoholic steatohepatitis (by liver biopsy and histological analysis within the last 6-months) and no other known or co-existing liver disease. An ultrasound was used to identify fatty liver disease. Another inclusion criterion was the willingness of the patient to participate in the study and the ability to tolerate the study protocol. 

Patients with other hepatic pathologies causing increased necroinflammatory activity were excluded from the study: chronic viral hepatitis was excluded using standard screening tests (HBs antigen for HBV and anti-HCV antibodies for HCV), alcohol-induced liver disease was established based on a history of alcohol abuse, evidenced by anamnesis (the estimated consumption of less than <20 g/day for women and <30 g/day for men were desired) and the CAGE questionnaire as well as indirect laboratory markers of alcohol consumption (an increased mean erythrocyte volume and isolated increase in gamma-glutamyl transpeptidase), hemochromatosis was excluded through iron saturation and liver biopsy, autoimmune hepatitis was excluded by antinuclear antibody titer, alpha-1-antitirpsin was measured to exclude deficiency, primary biliary cirrhosis was excluded through alkaline phosphatase levels (normal), anti-mitochondrial antibodies and liver biopsy, Wilson disease was excluded based on normal ceruloplasmin, hepatocellular carcinoma, and other neoplasms of the liver were excluded through imaging (ultrasound) and alfa-fetoprotein.

Drug-induced liver disease was also evaluated based on each patient’s prescriptions, and the patients with chronic treatment using known liver toxic drugs were excluded from the study. Furthermore, patients with drugs that could interfere with octanoate metabolism or that could cause NAFLD independent of the metabolic syndrome were also excluded: amiodarone, corticosteroids, methotrexate, stavudine, tetracycline, valproic acid, zidovudine. 

Comorbidities that excluded patients from the study included severe COPD (GOLD C or above), severe asthma, uncontrolled diabetes (HbA1C > 7%), severe congestive class failure (NYHA class 3 or above), and malabsorption syndromes.

Other exclusion criteria included pregnancy, hypersensitivity to ^13^C Sodium Octanoate, significant weight change during the study protocol (defined as >10%), a recent acute disease that required medical or surgical treatment (past 3 months), and patients participating in other clinical trials.

All patients underwent a comprehensive assessment, including medical history (including previous history, current treatments, alcohol consumption, and symptoms) and physical examination (including measurement of height, weight, waist circumference, blood pressure, heart rate, and abdominal palpation). The CAGE questionnaire was applied to each patient to investigate alcohol abuse. Height, weight, and waist circumference were measured. Body mass index (BMI) was calculated (kg/m^2^) as weight (kg) divided by height (m^2^), and the results were classified as underweight (<18), normal (18–25), overweight (25–29.0), and obese (degree 1—30–34.9, degree 2—35–39.9, degree 3 ≥40). Abdominal obesity was identified by measuring waist circumference (WC) at the midpoint between the lower border of the rib cage and the iliac crest. The blood pressure result was determined as the mean of the second and third readings of three consecutive blood pressure measurements. Each measurement was performed with the patient at rest for at least 10 min. 

The blood samples were obtained under fasting conditions and the tests performed include alanine aminotransferase (ALT), aspartate aminotransferase (AST), serum albumin, γ-glutamyl transpeptidase (GGT), total bilirubin, direct bilirubin, alkaline phosphatase, international normalized ratio (INR), glucose, cholesterol, including the high-density lipoprotein (HDL) and low-density lipoprotein (LDL) fractions, triglycerides, uric acid, serum concentration of iron and ferritin, urea, creatinine, HBs antigen, anti HCV antibodies, anti-mitochondrial antibodies, serum ceruloplasmin, alfa-fetoprotein, complete blood count, antinuclear antibody, alfa1-antitripsin, and HbA1c. The devices used to analyze the samples included CELL-DYN 370 (Abbot Diagnostics, IL, USA), ARCHITECT c8000 (Abbot Diagnostics, IL, USA), ACL TOP 500 (Instrumentation Laboratory, Bedford, MA, USA), Access 2 Immunoassay System (Beckman Coulter, Brea, CA, USA)), and the Dimension RXL analyzer (Siemens-Dade Behring, Erlangen, Germany).

All the patients were investigated by ultrasonography (US) using Acuson S2000 (Siemens AG, 91052 Erlangen, Germany). In our study, the quantification of hepatic steatosis was made using the classification of “bright liver”, which is based on a four-point scale of hyperechogenity: 0 = absent, 1 = light, 2 = moderate, 3 = severe, according to the difference between the densities of the liver and the right renal cortex. The spleen volume was estimated by a measurement of the spleen longitudinal diameter (SLD) using the maximum length obtained between the two poles of the spleen in postero-lateral scanning. The measurements were performed by a senior physician with 20 years of experience in abdominal ultrasound.

All the enrolled patients underwent histological assessment by percutaneous liver biopsy using the Menghini technique with a 1.4 mm diameter needle (Hepafix, Braun, Germany). The procedure was performed by a senior physician with expertise in liver biopsy. Specimen analysis was performed by an expert pathologist (20-year experience) blinded to the patient’s clinical results. The length of each liver biopsy was established, and the number of portal tracts was counted. Only liver fragments of at least 2.0 cm in length, which included eight portal tracts, were considered for histological assessment. The steatosis-activity-fibrosis (SAF) classification and fatty liver inhibition of the progression (FLIP) diagnosis algorithm were used to establish a positive diagnosis for non-alcoholic steatohepatitis (NASH) [12].

The ^13^C-Octanoate breath test is a non-invasive test proposed as an alternative to hepatic biopsy as well as other non-invasive tests. Breath tests require the patient to be administered a substrate marked with the carbon 13 stable isotope, which will then be metabolized, according to each substrate’s normal metabolism, excreted, and then measured. ^13^C Octanoate has physical and chemical properties that could make it an appropriate non-invasive marker for the quantification of hepatic mitochondrial β-oxidation by means of a breath test. Octanoate [CH_3_(CH_2_)_6_CO_2_H] is a medium-chain fatty acid that is readily absorbed from the intestine without incorporation into the micelles, and is rapidly transported to the liver through the portal venous system. In the hepatocytes, it is metabolized through β-oxidation in the mitochondria, resulting in acetyl coenzyme A (CoA). Furthermore, Acetyl CoA enters the Krebs cycle and is oxidized to CO_2_, which is then transported through the systemic circulation to the lungs and is eliminated in the breath. The exhaled ^13^CO_2_ is captured in specially designed breath test bags, which are then analyzed using a ^13^C/^12^C Infrared Spectrometer (IRIS Doc, Wagner). The ^13^C isotope was chosen as it is the only stable isotope of carbon, other than the ^12^C which is naturally present. ^13^C is not naturally present in the body, and to verify this, before the test, each patient is given the ^13^C substrate, and a trial measurement is made to confirm 0% of levels of exhaled ^13^CO_2_. Using carbon isotopes also presents significant advantages, as it is quickly eliminated through exhalation and easily measured using infrared spectrometry [13,14].

The breath test was performed after fasting overnight. The preprocedural ingestion of food and physical activity was prohibited. After collecting the control sample of the air into a bag, each subject ingested 100 mg of ^13^C octanoate labeled with stable non-radioactive isotope ^13^C (^13^C-Sodium Octanoate, Hanseaten-Apotheke, Germany) solved in 200 mL water. Breath samples were collected at the baseline and at 0, 15, 30, 45, 60, and 120 min from substrate administration. The analysis was performed at a maximum of 60 min from the last sample being collected. The parameters measured included ^13^CO_2_ exhaled and recovered as a function of time, which describes the velocity of substrate metabolization and ^13^CO_2_ exhalation—the percent recovery dose, measured in % per hour (PDR [%/h]) and cumulative exhaled ^13^CO_2_, which describes the total amount of substrate metabolized and ^13^CO_2_ exhaled—cumulative percent recovery dose, measured in the % of total dose (cPDR [%]). The analyzer performed all the measurements for a single patient simultaneously and calculated the PDR and cPDR automatically for each point in time.

A total number of 187 patients were investigated, out of which 82 were diagnosed with steatohepatitis and 64 were diagnosed with simple steatosis, and 41 were excluded due to exclusion criteria. The histological features were assessed using the SAF score, which includes the grade of steatosis, inflammatory activity, and fibrosis (Table 1). The severity of steatosis was assessed from S1 to S3. The activity of the disease, based on the evaluation of both hepatocyte ballooning and lobular inflammation, was classified as mild (A1), moderate (A2–A3), and severe (A4). We used clinically five fibrosis stages F0–F4. For the diagnosis of NASH, the histological features required were moderate activity (A ≥ 2), any stage of steatosis, and any stage of fibrosis.

A control group of 21 healthy volunteers was also investigated. The patients were divided into three groups. Group 1 included patients with steatohepatitis (liver biopsy result positive for steatohepatitis), group 2 included patients with simple steatosis (liver biopsy result negative for steatohepatitis), while group 3 included healthy volunteers (no liver biopsies were performed), with no evidence of liver disease.

The results were entered into a table using Microsoft Excel (Microsoft Corporation, Redmond, WA, USA) and analyzed using Excel (Microsoft Corporation), Python with the pandas, Statsmodels and SciPy libraries, and SPSS Version 23 (IBM Corporation). The normality of the variables was assessed using histograms and the Shapiro–Wilk test. In univariate analysis, the one-way ANOVA test was used to assess the statistical difference, and Tukey honestly significant difference post hoc test was used to analyze differences between the individual groups. The crosstabulation analysis for gender was performed using the Chi-Square test. Bivariate analysis was performed to assess the strength of the association between the breath test results and SAF score inflammation using Spearman’s rho test on patients with hepatic biopsies (groups 1 and 2). The receiver operating characteristic (ROC curve) was constructed for the best parameters, and an optimal cutoff was chosen using Youden’s J statistic method. Sensitivity, specificity, positive predictive, and negative predictive values were calculated. The overall accuracy was calculated using the area under the ROC curve (AUROC).

## 3. Results

### 3.1. Characteristics of Patients

The study group included a total number of 146 patients, out of which 82 were diagnosed with steatohepatitis (group 1), and 64 were diagnosed with simple steatosis—NAFL (group 2), while the control group included 21 healthy volunteers (group 3) (Figure 1). The patients were evenly distributed by age and gender, with no statistical difference between the groups. A total of 41 patients initially proposed for the study were excluded due to either positive chronic infection with HBV, HCV, or due to alcohol consumption. 

### 3.2. Univariate Analysis

The univariate analysis identifies several factors associated with non-alcoholic steatohepatitis, including aspartate and alanine aminotransferases, direct and indirect bilirubin, alkaline phosphatase, gamma-glutamyl transpeptidase, and triglycerides (Table 2).

Regarding the ^13^C Octanoate breath test, significant values were the PDR (%/h) at 15 min, 30 min, 45 min, 60 min, and 120 min from substrate administration (Figure 2a), as well as the cPDR at 30 min, 45 min, 60 min, and 120 min from substrate administration (Figure 2b).

Post hoc tests show that the differences observed were between group 1 vs. group 2 and between group 1 vs. group 3. There was no statistical difference between groups 2 and 3 (Table 3).

### 3.3. Bivariate Analysis

The bivariate analysis identifies a strong positive correlation between PDR and the activity score, with the strongest being for the value measured at 30 min—r = 0.65. Good correlations were also observed for cumulative doses measured later than 45 min from substrate administration, with the strongest correlation appearing at 120 min—r = 0.69 (Table 4).

### 3.4. Diagnostic Performance of the ^13^C Breath Test

Patients with steatohepatitis had significantly increased exhaled ^13^CO_2_ during the test. Both the PDR at 15 min and the cumulative dose at 120 min were good parameters in evaluating steatohepatitis in both univariate and bivariate analysis. The area under the receiving operator characteristic (ROC curve) was 0.902 for the diagnosis of PDR at 15 min (Figure 3a). Based on the ROC curve, a cutoff value of 17.14 was chosen, which yielded a sensitivity of 95%, a specificity of 74%, a positive predictive value of 78%, and a negative predictive value of 94%. The cumulative recovered dose at 120 min had an area under the receiving operator characteristic (ROC curve) of 0.899 for diagnosis (Figure 3b). A cutoff value of 29.02 was chosen, which yielded a sensitivity of 81%, a specificity of 86%, a positive predictive value of 85%, and a negative predictive value of 83% (Table 5).

## 4. Discussion

Metabolic-associated fatty liver disease (MAFLD) is the most common liver disorder in many countries, where the major risk factors for MAFLD—the components of metabolic syndrome—are frequent [15]. NAFLD is subdivided at least into two histologically distinguishable entities: nonalcoholic fatty liver (NAFL) and nonalcoholic steatohepatitis (NASH), with different prognoses [16]. Out of the patients with NAFLD, only a minority have NASH, the progressive disease with severe liver outcomes. So far, invasive liver biopsy is the only method that can differentiate patients with NASH from those with simple steatosis (NAFL), but the disadvantages of liver biopsy consist of the sampling error, inter-observer disagreement, risks and complications, and the fact that it is a snapshot of a dynamic and ever-changing process. The replacement of liver biopsy in the assessment of chronic liver disease is the goal of any non-invasive technique.

One hallmark of nonalcoholic steatohepatitis is mitochondrial dysfunction with consequences on hepatocyte bioenergetics, reactive oxygen species (ROS) homeostasis, inflammation, and cell death [17,18].

The evaluation of liver function usually supposes a combination of clinical and serum parameters into scores limited by their „static” character in the predicting of outcomes.

‘Dynamic’ liver tests evaluating metabolic activity over a period of time are most reliable in predicting the severity of liver disease according to the functional reserve of the liver and in predicting survival [19,20].

In this context, we decided to evaluate the role of the ^13^C Octanoate breath test (^13^C-OBT), as a surrogate biomarker of NASH, in a cohort of patients with histologically proven NAFLD and to report our single-center experience. 

Our study showed that ^13^C-OBT had a good efficacy for identifying patients with NASH from those with NAFL (simple steatosis). By examining the speed of octanoic acid metabolism, the patients with NASH had a significant increase in the ^13^CO_2_ exhalation rate at each time point (PDR) compared with patients with NAFL and the control group. Regarding the metabolism capacity in contrast to the controls and patients with NAFL, patients with NASH had a higher capacity to metabolize octanoic acid, expressed as the cumulative percentage of recovery of ^13^CO_2_ in the breath (cPDR). The highest difference between the patients with NASH and those with steatosis was noted for cPDR at 120 min after octanoic acid ingestion (31.9 ± 3.35 vs. 25.3 ± 3.52, *p* < 0.001). The characteristic curves regarding the speed and the metabolism capacity of octanoic acid allowed for differentiation between, on the one hand, the patients with NASH and, on the other hand, the patients with NAFL and the control group. Our study also showed that there were no significant differences between the patients with NAFL and healthy subjects according to the octanoic acid metabolism.

To evaluate the performance of OBT in distinguishing patients with NASH from those with NAFL, we used the area under the ROC curve. The best diagnostic power for NASH was found for the PDR at 15 min; using a cut-off value of 17.14%, the AUROC was very good, 0.902, with good sensitivity of 95% and specificity of 74% and especially with an excellent negative predictive value of 94%. The cumulative recovered dose (cPDR) at 120 min had an AUROC of 0.899 for the diagnosis of NASH for a cut-off value of 29.02%, with a well-balanced sensitivity of 81% and specificity of 86% 

In previous studies, there have been conflicting results regarding the fatty acid oxidation met in NAFLD. Some studies conclude that hepatic mitochondrial oxidation is increased in NASH [21,22], but another study demonstrated no differences between patients with MAFLD and healthy controls [23].

There are few studies that used ^13^C-OBT as a surrogate marker to estimate mitochondrial fatty acid oxidation. One of these was based on an experimental model (rat) and emphasized the good sensitivity of OBT in evaluating mitochondrial liver function in acute hepatitis and toxic-induced cirrhosis [24]. However, the results of the ^13^C-Octanoate breath test in patients with NAFLD have been different. In a study that explored the metabolic mitochondrial pathway in NAFLD, using the same protocol as us, Miele et al. reported a significant increase in the mitochondrial oxidation of patients with NASH compared with the healthy controls. The cumulative percentage of ^13^C-Octanoate oxidation over 120 min in patients with NASH was comparable with our results: (33.6 ± 4.6% vs. 32.1 ± 3.6%) [25].

Another study that used ^13^C-OBT to diagnose the severity of hepatic injury in patients with NAFLD also showed that this test distinguished between patients with NAFL and those with NASH; the percentage of the ^13^C dose recovered was the best parameter, as in our results. The authors reported a highly significant correlation between OBT and abnormal values for insulin levels and the homeostatic model assessment of insulin resistance (HOMA), both being specific features of NASH [26].

However, there are studies where the octanoic acid metabolism did not differ among patients with NASH and the healthy controls [27] and even more, without differences detected, regardless of the presence and severity of cirrhosis [28].

Among other breath tests used for assessing liver mitochondrial function is the ^13^C-ketoisocaproate test (KIKA). This substrate uses another mitochondrial metabolic pathway than octanoic acid. In a study performed on patients with MAFLD, those with NASH had a significantly lower mitochondrial decarboxylation capacity compared with healthy subjects. The mitochondrial decarboxylation capacity was inversely correlated with the severity of the lesions in NASH. The ^13^CO_2_ exhalation rate secondary to KIKA metabolism was lower in patients with an advanced stage of NASH, and the test was proposed for disease progression follow-up [29].

Methionine is another substrate of the breath test, which is used to evaluate the mitochondrial decarboxylation of a metabolic pathway and is more sensitive for the detection of the early stages of NASH. The authors concluded that a ^13^C Methionine-breath test is a tool useful to differentiate patients with NASH from those with steatosis and to monitor the disease progression [30].

Although all these substrates demonstrated a good ability to identify patients with NASH, octanoic acid has the advantage for exploring the most important mitochondrial metabolic pathway, the ß-oxidation of fatty acid, and the main source of oxidative stress [31]. Fatty acid oxidation is increased in patients with NASH secondary to the higher mitochondrial biogenesis and mitochondrial mass than in the liver tissue of healthy individuals. The increased mitochondrial oxidation of fat increases the delivery of electrons to the respiratory chain; some of these electrons react directly with oxygen to form ROS [32]. This scenario can explain our results regarding the higher values of the dynamic of octanoic acid metabolism characteristic in patients with NASH, distinguishing them from the large cohort of NAFLD patients. 

Sustained mitochondrial oxidative flux results in increased ROS production followed by DNA damage, mitochondrial structural abnormalities, lipid peroxidation, and hepatocyte death. Mitochondrial ROS trigger proinflammatory cytokines such as interleukin 6, tumor necrosis factor α, and interleukin 1, which are crucial mediators of inflammation in NASH [17,33,34]. According to these pathogenic features, our study showed a strong positive correlation between PDR and activity score—r = 0.65.

As NASH progress to advanced liver disease, mitochondrial adaptation and flexibility to the higher metabolic demands in NASH become compromised and lead to mitochondrial disfunction, bioenergetic failure, and cellular apoptosis. All these changes can explain the decreased values of OBT metabolism as the disease progresses and generates conflicting results. 

Taking into account all these pathogenic steps in NASH, we consider that OBT may have a clinical utility in patients with steatosis imaging detected to identify patients at risk for NASH, especially “fast progressors” [35], who advance to cirrhosis within 10 years and also to monitor all NAFLD patients for disease progression to advanced liver disease [36].

Our study has some limitations: primarily the fact that this is a single-center study that comprised a small number of patients. The liver biopsy that served as a reference was performed before breath tests, sometimes as long as 6-months before. Selection bias is another limitation: we excluded patients with any clinical liver decompensation, which could influence the cutoff values, sensitivity, and specificity in distinguishing fibrosis stages. 

Nevertheless, the special advantages of non-invasive breath tests are safety, simplicity of the protocol, the ability to provide immediate results, and high patient acceptance.

## 5. Conclusions

The ^13^C-Octanoate breath test is a promising method in differentiating patients with NASH from those with simple steatosis (NAFL). The patients with NAFL have the same results regarding this test as the healthy controls. The ^13^C-Octanoate breath tests have the potential to become tools in clinical practice as methods for the diagnosis and follow-up of patients with NAFLD, thus decreasing the need for liver biopsy. 

Further investigation of these test results may also benefit from analysis with artificial intelligence models, which would allow the correlation of a large number of other parameters [37]

## Figures and Tables

**Figure 1 diagnostics-12-02935-f001:**
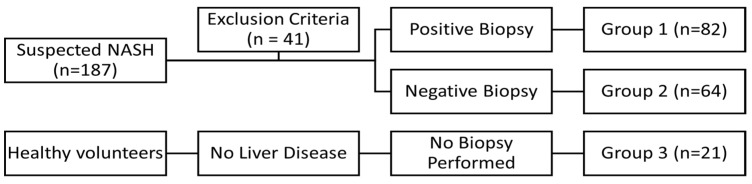
Study Design.

**Figure 2 diagnostics-12-02935-f002:**
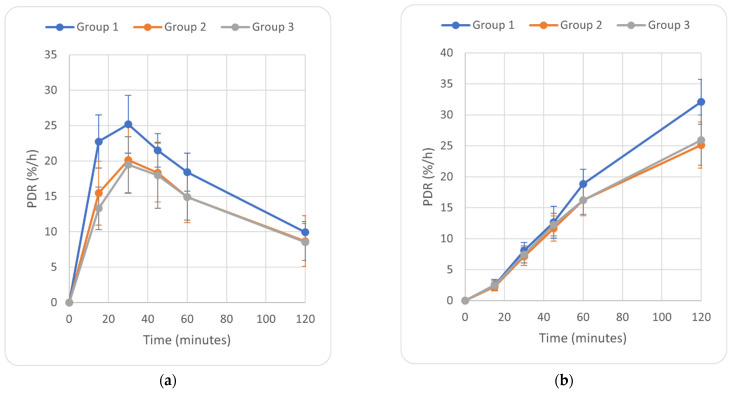
(**a**) PDR measurements; (**b**) cPDR measurements.

**Figure 3 diagnostics-12-02935-f003:**
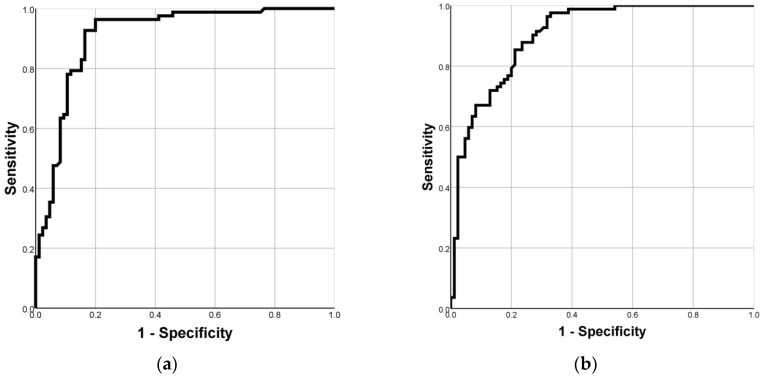
(**a**) ROC curve for PDR at 15 min; (**b**) ROC curve for cPDR at 120 min.

**Table 1 diagnostics-12-02935-t001:** Histopathological characteristics. SAF score.

	*0*	*1*	*2*	*3*	*4*
*Steatosis*	0	43	53	50	0
*Activity*	60	4	19	48	15
*Fibrosis*	63	66	15	2	0

**Table 2 diagnostics-12-02935-t002:** Univariate Analysis.

Variable	Group 1	Group 2	Group 3	Method	df	*p*-Value
Number	82	64	21			
Sex (M/F)	42 M|40 F	28 M|36 F	8 M|13 F	Chi Square	2	0.467
Age (years)	51.09 (SD = 12.03)	48.31 (SD = 12.49)	48.67 (SD = 7.49)	ANOVA	2	0.337
Blood glucose (mg/dL)	116.2 (SD = 37.14)	121.94 (SD = 47.51)	119.48 (SD = 36.67)	KW	2	0.504
Urea (mg/dL)	34.73 (SD = 8.09)	34.05 (SD = 12.67)	39.05 (SD = 15.78)	KW	2	0.181
Creatinine (mg/dL)	0.79 (SD = 0.36)	0.9 (SD = 0.19)	0.86 (SD = 0.16)	KW	2	0.737
Total Bilirubin (mg/dL)	1.11 (SD = 0.32)	0.65 (SD = 0.59)	0.59 (SD = 0.62)	KW	2	<0.001 *
Direct Bilirubin (mg/dL)	0.34 (SD = 0.15)	0.16 (SD = 0.16)	0.13 (SD = 0.13)	KW	2	<0.001 *
ALP (mg/dL)	86.24 (SD = 20.28)	75.27 (SD = 20.76)	74.57 (SD = 21.29)	KW	2	0.047 *
GGT (U/L)	169.39 (SD = 163.87)	71.75 (SD = 63.24)	60.1 (SD = 60.11)	KW	2	0.004 *
Cholesterol (mg/dL)	188.13 (SD = 38.15)	199.3 (SD = 36.16)	202.86 (SD = 37.08)	KW	2	<0.001 *
HDL Cholesterol (mg/dL)	39.73 (SD = 15.66)	48.95 (SD = 19.67)	48.05 (SD = 16.13)	KW	2	<0.001 *
LDL Cholesterol (mg/dL)	126.53 (SD = 29.12)	111.53 (SD = 23.71)	119.51 (SD = 25.82)	KW	2	<0.001 *
Triglycerides (mg/dL)	113.05 (SD = 50.05)	208.45 (SD = 76.73)	199.52 (SD = 63.22)	KW	2	<0.001 *
AST (U/L)	115.49 (SD = 110.09)	25.92 (SD = 7.65)	26.62 (SD = 6.07)	KW	2	<0.001 *
ALT (U/L)	159.35 (SD = 157.67)	44.42 (SD = 12.38)	46.95 (SD = 14.45)	KW	2	<0.001 *
PDR at 15 min (%/h)	22.75 (SD = 3.78)	15.45 (SD = 4.5)	13.3 (SD = 3.02)	KW	2	0.072 *
PDR at 30 min (%/h)	25.2 (SD = 4.09)	20.14 (SD = 4.7)	19.47 (SD = 3.97)	KW	2	<0.001 *
PDR at 45 min (%/h)	21.5 (SD = 2.36)	18.35 (SD = 4.16)	17.98 (SD = 4.68)	KW	2	<0.001 *
PDR at 60 min (%/h)	18.42 (SD = 2.7)	14.91 (SD = 3.6)	14.9 (SD = 3.23)	KW	2	<0.001 *
PDR at 120 min (%/h)	9.92 (SD = 1.52)	8.68 (SD = 3.56)	8.54 (SD = 2.6)	KW	2	0.001 *
cPDR at 15 min (%)	2.52 (SD = 0.91)	2.24 (SD = 0.68)	2.46 (SD = 0.76)	KW	2	0.212
cPDR at 30 min (%)	8.1 (SD = 1.29)	7.12 (SD = 1.44)	7.45 (SD = 1.38)	KW	2	0.031 *
cPDR at 45 min (%)	12.65 (SD = 2.56)	11.63 (SD = 2.02)	12.28 (SD = 1.84)	KW	2	<0.001 *
cPDR at 60 min (%)	18.86 (SD = 2.34)	16.22 (SD = 2.5)	16.23 (SD = 2.33)	KW	2	<0.001 *
cPDR at 120 min (%)	32.11 (SD = 3.62)	25.11 (SD = 3.71)	25.93 (SD = 4.06)	KW	2	<0.001 *

SD = standard deviation, AST = aspartate aminotransferase, ALT = alanine aminotransferase, HDL = high density lipoprotein, LDL = low density lipoprotein, ALP—alkaline phosphatase, GGT = gamma glutamyl transpeptidase, cPDR = cumulative dose, ANOVA = one-way analysis of variance, KW = independent samples Kruskal–Wallis, df = degrees of freedom *—Statistically significant at the 0.05 level.

**Table 3 diagnostics-12-02935-t003:** Univariate post hoc tests.

Variable	*p*-Value (Group 1 vs. Group 2)	*p*-Value (Group 1 vs. Group 3)	*p*-Value (Group 2 vs. Group 3)	Method
Total Bilirubin (mg/dL)	<0.001 *	<0.001 *	0.968	Dunn
Direct Bilirubin (mg/dL)	<0.001 *	<0.001 *	0.722	Dunn
ALP (mg/dL)	<0.001 *	0.050	0.900	Dunn
GGT (U/L)	<0.001 *	<0.001 *	0.900	Dunn
Cholesterol (mg/dL)	<0.001 *	<0.001 *	0.520	Dunn
HDL Cholesterol (mg/dL)	<0.001 *	0.120	0.900	Dunn
LDL Cholesterol (mg/dL)	<0.001 *	0.547	0.460	Dunn
Triglycerides (mg/dL)	<0.001 *	<0.001 *	0.830	Dunn
AST (U/L)	<0.001 *	<0.001 *	0.961	Dunn
ALT (U/L)	<0.001 *	<0.001 *	0.662	Dunn
PDR at 15 min (%/h)	<0.001 *	0.008 *	0.961	Dunn
PDR at 30 min (%/h)	<0.001 *	<0.001 *	0.810	Dunn
PDR at 45 min (%/h)	<0.001 *	<0.001 *	0.900	Dunn
PDR at 60 min (%/h)	<0.001 *	<0.001 *	0.900	Dunn
PDR at 120 min (%/h)	0.010 *	0.082	0.900	Dunn
cPDR at 15 min (%)	0.090	0.900	0.521	Dunn
cPDR at 30 min (%)	<0.001 *	0.125	0.600	Dunn
cPDR at 45 min (%)	0.021 *	0.782	0.500	Dunn
cPDR at 60 min (%)	<0.001 *	<0.001 *	0.900	Dunn
cPDR at 120 min (%)	<0.001 *	<0.001 *	0.664	Dunn

AST = aspartate aminotransferase, ALT = alanine aminotransferase, HDL = high density lipoprotein, LDL = low density lipoprotein, ALP—alkaline phosphatase, GGT = gamma glutamyl transpeptidase, cPDR = cumulative dose, *—Statistically significant at the 0.05 level.

**Table 4 diagnostics-12-02935-t004:** Bivariate analysis (Spearman’s Rho) with the SAF parameters.

	Steatosis (S)	Inflammatory Activity (A)	Fibrosis (F)
Age	−0.01	0.01	0.17
Blood glucose	0.03	−0.05	−0.03
Urea	−0.15	0.05	0.10
Creatinine	−0.09	−0.14	−0.06
Total Bilirubin	0.20	0.56	0.41
Direct Bilirubin	0.10	0.56	0.38
ALP	−0.07	0.22	0.15
GGT	0.06	0.17	0.18
Cholesterol	−0.12	−0.16	−0.17
HDL Cholesterol	−0.06	−0.16	−0.13
LDL Cholesterol	−0.04	0.15	0.11
Triglycerides	0.02	−0.58	−0.43
AST	−0.06	0.64	0.40
ALT	−0.08	0.64	0.41
PDR at 15 min	0.09	0.65	0.40
PDR at 30 min	−0.01	0.47	0.35
PDR at 45 min	0.03	0.39	0.32
PDR at 60 min	0.06	0.46	0.33
PDR at 120 min	0.04	0.32	0.28
cPDR at 15 min	−0.09	0.13	0.06
cPDR at 30 min	−0.07	0.16	0.11
cPDR at 45 min	0.00	0.39	0.27
cPDR at 60 min	−0.15	0.31	0.22
cPDR at 120 min	0.08	0.69	0.46

AST = aspartate aminotransferase, ALT = alanine aminotransferase, HDL = high density lipoprotein, LDL = low density lipoprotein, ALP—alkaline phosphatase, GGT = gamma glutamyl transpeptidase, cPDR = cumulative dose.

**Table 5 diagnostics-12-02935-t005:** Diagnostic accuracy for Octanoate breath test.

Variable	AUROC	95% CI Interval		Cutoff	Sensitivity	Specificity	PPV	NPV
PDR at 15 min	0.902	0.855	0.949	17.14	95%	74%	78%	94%
PDR at 30 min	0.811	0.746	0.876	20.85	86%	66%	71%	84%
PDR at 45 min	0.735	0.656	0.813	17.99	95%	53%	66%	92%
PDR at 60 min	0.788	0.716	0.860	15.86	86%	71%	73%	85%
PDR at 120 min	0.667	0.581	0.753	8.46	83%	54%	63%	77%
cPDR at 15 min	0.563	0.475	0.652	3.31	21%	98%	89%	56%
cPDR at 30 min	0.681	0.601	0.762	7.83	63%	68%	65%	66%
cPDR at 45 min	0.592	0.505	0.679	13.69	36%	82%	66%	57%
cPDR at 60 min	0.780	0.710	0.849	16.05	91%	55%	66%	87%
cPDR at 120 min	0.899	0.853	0.945	29.02	81%	86%	85%	83%

cPDR = cumulative dose, AUROC = area under receiver operating characteristic, 95% CI = 95%, confidence, PPV = positive predictive value, NPV = negative predictive value.

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
