# Peer review of "The Role of Noninvasive 13C-Octanoate Breath Test in Assessing the Diagnosis of Nonalcoholic Steatohepatitis"

_diagnostics, 2022, doi:10.3390/diagnostics12122935_

Round 1

Reviewer 1 Report

It is a very interesting study.

The following are suggestions to improve th emanuscript"

1. The Introduction needs to better explaining why this particular isotope was preferred among others for which data is available

2. The same should be elucidated in the Methods section too

3. The following references maybe added and discussed:

a. Molina-Molina, E., Shanmugam, H., Di Palo, D., Grattagliano, I., Portincasa, P. (2021). Exploring Liver Mitochondrial Function by 13C-Stable Isotope Breath Tests: Implications in Clinical Biochemistry. In: Palmeira, C.M., Rolo, A.P. (eds) Mitochondrial Regulation. Methods in Molecular Biology, vol 2310. Humana, New York, NY. https://doi.org/10.1007/978-1-0716-1433-4_11

b. Furnari Manuele, Savarino Vincenzo and Giannini G. Edoardo, Use of Liver Breath Tests to Assess Severity of Nonalcoholic Fatty Liver Disease, Reviews on Recent Clinical Trials 2014; 9(3) . https://dx.doi.org/10.2174/1574887109666141216104917

4. The conclusion should mention the way forward as it is an evolving modality 

Author Response

Dear Sir/Madam,

I would like to thank you for your review of our original article.

Regarding your questions:

  1. The difference between negative and positive biopsy is narrow. Can you explain this notice?

We had a lot of patients with positive biopsies, mainly because a significant number were specifically referred to our center for further care (with a high degree suspicion for NASH). Patients with suspected or histologically confirmed NASH were also more likely to accept the participation in the study.

  1. What about occult HBV? Did you investigate patients for HB core antibody?

We did not test for occult HBV.

  1. Please add a degree of freedom for each p-value.

Table 2 was modified accordingly. Please note that df in Kruskal Wallis and Chi Square is groups -1. For ANOVA the between groups df was used. The within groups df for ANOVA test was 164.

Sincerely ,                                            

Prof. Univ. Dr. Carmen Fierbinteanu-Braticevici                      

Reviewer 2 Report

This is a nice article that discusses a hot issue. Some comments are mentioned below:

1- the difference between negative and positive biopsy is narrow. Can you explain this notice?

2- what about occult HBV? Did you investigate patients for HB core antibody?

3-please add a degree of freedom for each p-value

Author Response

Dear Sir/Madam,

I would like to thank you for your review of our original article.

Regarding your questions:

  1. The introduction needs to better explain why this particular isotope was preferred among other for which data is available.

Clarifications were added in the introduction as follows:

The 13C isotope was chosen as it is the only stable isotope of carbon other than 12C.

  1. The same should be elucidated in the Methods section too.

Clarifications were added in the Methods section as follows:

The 13C isotope was chosen as it is the only stable isotope of carbon, other than the 12C which is naturally present. 13C is not naturally present in the body, and to confirm this, before each patient is given the 13C substrate, a trial measurement is made, that confirms 0% levels of exhaled 13CO2. Using carbon isotopes also presents significant advantages, as it is quickly eliminated through exhalation and easily measured using infrared spectrometry.

  1. The following references may be added and discussed:

References were added in the text accordingly.

  1. The conclusion should mention the way forward as it is an evolving modality

Conclusions were updated:

Further investigation of these teste results may also benefit from analysis with artificial intelligence, which would allow for correlations with a large number of other parameters.

Sincerely,

Prof. Univ. Dr. Carmen Fierbinteanu-Braticevici